# Social Determinants of Rural Health Workforce Retention: A Scoping Review

**DOI:** 10.3390/ijerph16030314

**Published:** 2019-01-24

**Authors:** Catherine Cosgrave, Christina Malatzky, Judy Gillespie

**Affiliations:** 1Department of Rural Health, Faculty of Medicine, Dentistry and Health Sciences, University of Melbourne, Docker St Wangaratta, 3677 Victoria, Australia; 2School of Public Health and Social Work, Faculty of Health, Queensland University of Technology, Kelvin Grove campus, Brisbane, 4059 Queensland, Australia; christina.malatzky@unimelb.edu.au; 3School of Social Work, Faculty of Health and Social Development, The University of British Columbia, Okanagan campus, Kelowna, BC V1V 1V7, Canada; judy.gillespie@ubc.ca

**Keywords:** rural health, workforce, retention, turnover, allied health, nursing, medical professionals, social processes, rural place, scoping review

## Abstract

Residents of rural and remote Australia have poorer health outcomes than their metropolitan counterparts. A major contributor to these health disparities is chronic and severe health workforce shortages outside of metropolitan areas—a global phenomenon. Despite emerging recognition of the important influence of place-based social processes on retention, much of the political attention and research is directed elsewhere. A structured scoping review was undertaken to describe the range of research addressing the influence of place-based social processes on turnover or retention of rural health professionals, to identify current gaps in the literature, and to formulate a guide for future rural health workforce retention research. A systematic search of the literature was performed. In total, 21 articles were included, and a thematic analysis was undertaken. The themes identified were (1) rural familiarity and/or interest, (2) social connection and place integration, (3) community participation and satisfaction, and (4) fulfillment of life aspirations. Findings suggest place-based social processes affect and influence the retention of rural health workforces. However, these processes are not well understood. Thus, research is urgently needed to build robust understandings of the social determinants of rural workforce retention. It is contended that future research needs to identify which place-based social processes are amenable to change.

## 1. Introduction

Residents of rural and remote Australia have poorer health outcomes than their metropolitan counterparts [1]. Compared to metropolitan-based residents, those in rural and/or remote areas of Australia live shorter lives, acquire greater levels of chronic disease, sustain more injuries, experience poorer mental health, and have less access to and use of health services [1]. Given that a high proportion of Indigenous Australians live in rural and remote Australia, health disparities between metropolitan and rurally based residents also reflect health inequities between Indigenous and non-Indigenous Australians [2]. A major contributor to reduced access to and use of health services by rural residents in particular is a chronic and severe health workforce shortage outside of metropolitan Australia [3,4]. High turnover among rural health professionals was found to have considerable direct and indirect effects on rural health services and community health outcomes [5,6]. Attracting and retaining experienced and skilled health professionals to work and live in rural areas is a global issue and a major policy focus of many governments [7,8]. In countries that have large land masses with small, broadly dispersed rural and remote populations, such as Australia and Canada, there are additional challenges in providing adequately staffed rural health services. There is also greater reliance on the public health sector as there is limited access to private health services [9]. These rural communities also have a higher proportion of Indigenous peoples who already experience discrimination and disadvantage in the health system [10,11].

In Australia, the federal government invested in policies and strategies aimed at addressing rural health workforce shortages since early 2000 [12]. These policy efforts include increasing the number of government-funded university training places for health students, setting quotas in university health courses for students from rural backgrounds, and offering financial incentives and supports to encourage qualified health professionals to “go rural” [12]. However, until recently, political attention primarily focused on the medical profession. Medical students from a rural background are purposively selected and a proportion are required to complete an extended rural training year or undertake all their training rurally. These strategies were somewhat effective in increasing the supply of doctors working in rural Australia, especially in primary care settings [12]. There was also some improvement in the overall size of Australia’s rural health workforce. However, serious staffing shortages persist, particularly in allied health and nursing [12,13]. For the allied health workforce, maldistribution is a contributing factor, with an oversupply in metropolitan areas and an undersupply in rural areas, especially of experienced staff, which intensifies with degree of remoteness [5,14]. Having a fully staffed and adequately skilled rural allied health workforce is essential for the management of high and increasing levels of chronic health conditions and mental illness (including increased suicide risk) experienced by rural residents [15,16]. Australia’s nursing workforce is aging and, as the baby boomer generation of nurses retires over the next ten years, significant workforce shortages across the board are forecast [17,18]. These shortages are likely to impact heavily on the rural health workforce as nurses represent the largest component of Australia’s overall health workforce. Nurses are also the mainstay of health service provision in rural communities, and even more so in remote communities where they are often the only on-the-ground health professionals [19].

Research focused on the recruitment and retention of the rural health workforce in advanced economies identified the decisions of health professionals to stay or leave rural practice to be informed by a range of issues including job satisfaction, professional development, and career advancement opportunities, which intersect in unique ways [20,21]. Some health workforce studies investigating recruitment and retention categorize impacting issues into “push” (to leave the job) or “pull” (to take a job or stay in the job) factors [22,23]. To date, the majority of rural health recruitment and retention research focused on recruitment influences and the medical workforce. From this body of research, there is strong evidence that rural background, extended rural placements, and cultural immersion all positively influence the rural practice intentions of medical graduates [24,25]. However, given the different educational pathways and incentives offered to attract and retain medical practitioners to work rurally in comparison to those for allied health and nursing, the transferability or relevance of these findings beyond the medical field is questionable. The more limited body of nursing and allied health workforce rural retention research mostly focused on single professions and job satisfaction and/or professional and career development opportunities. The social dimensions of retention are often ignored altogether or identified with little explanation or exploration of how they operate and influence retention. However, there is an emerging body of research identifying and investigating the influences of place-based social processes on rural allied health and nursing workforce retention [26,27,28]. By the term “social processes”, the authors are referring to human practices and actions involving interactions that are essential to the formation of social relationships [29,30,31]. From the perspectives of the authors, social interactions are influenced by and influence the specific contexts or “places” in which they occur [32,33,34,35]. An emerging body of research holds great potential for progressing the field of rural workforce retention more broadly because place-based social processes impact upon the individual regardless of specific profession.

A better understanding is required of the ways in which place-based social processes manifest or are relevant for health professionals living and working in their towns of origin compared to those relocating from elsewhere. The main aim of this paper was, thus, threefold. Firstly, this paper aims to give readers a clear understanding of current knowledge on this topic by providing a comprehensive synthesis and analysis of existing rural allied health, nursing, and medical workforce literature that addresses the influence of place-based social processes on retention. Secondly, as part of the analysis, this paper aims to identify current research gaps in this field. Thirdly, in considering both existing knowledge and gaps in the literature, the paper aims to provide an outline of how research in this field could be developed to build the evidence base needed to effect change in the retention of a rural health workforce.

## 2. Materials and Methods

A structured scoping review was undertaken to describe the range of research addressing the influence of place-based social processes on turnover or retention of rural health professionals, to identify current gaps in the literature, and to formulate a potential guide for future research in this area. Undertaking a scoping review was considered appropriate for this research topic because it was not previously comprehensively reviewed [36]. In a scoping review, the quality of the studies selected is not assessed. Otherwise, a scoping review is similar to undertaking a systematic review and each stage of the process is undertaken in a rigorous and transparent manner. The five-stage framework proposed by Arksey and O’Malley was used to guide the conduct and reporting of this scoping review [36]. These stages are (1) identifying the research question, (2) identifying relevant studies, (3) study selection, (4) charting the data, and (5) collating, summarizing, and reporting the results [36]. The research question guiding this scoping review was two-pronged. Firstly, what are the types of place-based social processes impacting turnover and/or retention of rural health professionals identified in existing literature? Secondly, what is currently known through research about how these processes influence retention? To collate, summarize, synthesize, and report on the literature identified, a thematic analysis was undertaken. Other than an assessment of evidence strength, the scoping review complies with the PRISMA Statement [37].

### 2.1. Definitions of Key Terms

Retention refers to the length of time between commencement and cessation of employment with a particular employer [38,39]. Turnover is commonly used to gauge workforce retention and is a measure of staff exits in a specified time period. Given the challenges of recruiting former staff into service-based research, studying turnover can be difficult. However, researching turnover intention—an individual’s thoughts about leaving and intention to resign—among current staff is often used as a proxy measure of turnover because it is a strong predictor of individuals’ decisions to actually leave a position [40]. In this paper, the terms “turnover intention”, “turnover”, “retention”, and “reasons for staying/leaving” are used interchangeably. Given the strength of existing evidence that retention challenges increase with remoteness [41], all areas outside major cities are of interest in rural retention research. Herein, the use of the term “rural” should be considered to also include regional and remote unless otherwise specified. The term rural health workforce or rural health professionals should be considered to include allied health, nursing, and medical professionals unless specified otherwise. To differentiate between health professionals living and working in their town of origin and those who relocated from elsewhere, the terms “locals” and “non-locals” are used.

### 2.2. Identifying Relevant Studies

The search was conducted in June 2018 using database-specific search strings in MEDLINE (Ovid), Web of Science, Embase Classic, CINAHL, PsycINFO, and Informit Health Collection. The terms searched were “community attach*” OR “community satisfaction” OR “social network*” OR “social bond*” OR “sense of belonging” OR “sense of community” OR “place attach*” AND “retention” OR “turnover” OR “leave” OR “remain” OR “retain” AND “rural*” OR “remote*” OR “regional*” AND “allied health*” OR “dentist*” OR “dietetic*” OR “dietician*” OR “exercise physiologis*” OR “medic*” OR “medical laboratory technician*” OR “medical technolog*” OR “nurs*” OR “occupational therap*” OR “patholog*” OR “pharmac*” OR “physiologis*” OR “physiotherap*” OR “podiatr*” OR “psycholog*” OR “radiograph*”. Citation snowballing from the reference lists of the included articles was also undertaken [42].

### 2.3. Study Selection: Inclusion and Exclusion Criteria

Articles included concerned rural health workforce turnover or retention studies (in allied health, nursing, and/or medical fields) that identified place-based social processes as significant determinants and offered an explanation as to how they influence turnover or retention. Studies included empirical research and evaluations, including systematic reviews. During the initial search, two authors (C.C. and C.M.) reviewed the identified articles and independently screened titles and abstracts against pre-determined exclusion criteria: (i) language other than English, (ii) publication prior to 1995, and (iii) not based on findings from Australia, Canada, New Zealand, or the United States of America (USA).

## 3. Results

In total, 21 peer-reviewed journal articles meeting the inclusion criteria were identified. The results in each stage of the search and screening processes are illustrated in Figure 1. The characteristics of the included articles are summarized in Table 1. In all of the 21 articles, place-based social processes were found to have either the most or a major influence on health professionals’ decisions to stay in or leave a rural position. The thematic analysis identified a process of integration and adaptation involving four place-based social processes: (1) rural familiarity and/or interest, (2) social connection and place integration, (3) community participation and satisfaction, and (4) fulfillment of life aspirations. The articles were synthesized and discussed under the thematic sub-headings below. A summary of the themes, related categories, and supporting literature is provided in Table 2. For details pertaining to methods employed/type of study reported on, study population (and, where applicable, response rate), article’s focus, and relevant findings of each included article, see Table 3.

### 3.1. Rural Familiarity and/or Interest

Health professionals from rural backgrounds identify the existence of strong social bonds, familiarity with the physical environment, and enjoyment of a rural lifestyle as all having strong “pull effects” on their decision to take up or stay in rural positions [43,44,45,46,47,48,49]. In a study of 43 new graduate allied health professionals working in rural Victoria, Australia, existing social supports were described by participants as being significant for taking up a position close to “home” [43]. In Hancock et al.’s study of physicians working in rural settings in the USA, rural backgrounds were described as “priming” participants to choose rural practice [46]. In this study, seven of the 22 participants described having chosen rural practice because they wanted to live in a familiar natural or social environment that supported their feelings of “at-home-ness” [46]. This resulted in physician participants either choosing to work in the town they were raised in or one close by, or in another town with a similar population size and physical environment [46]. Hancock et al. also argued that rural “priming” extended beyond persons from a rural background and also influenced those from non-rural backgrounds who, as children or young adults, had positive rural recreation or employment experiences. Interestingly, however, Hancock et al. found that, while 14 of their 22 participants had rural experiences as part of their undergraduate or postgraduate medical education, only one participant identified this experience as influencing their decision to practice in rural USA. Similarly, Kulig et al.’s Canadian study identified two categories of nurses working in rural areas: those who were “going home” because of their attachment to community and satisfaction with the lifestyle offered, and those for whom the rural town was “becoming home” [50]. In this latter group, participants had either followed a spouse, had partnered with a person living in the town, or moved to the town for work and consequently decided to make it their home. Kulig et al. found that non-local participants who actively chose a rural town as their home had an adventurous temperament and, once relocated, were attracted to the town’s physical environment and lifestyle. These themes of rural as “home” and rural lifestyle preference were also examined in Gillespie and Redivo’s study of child and youth mental health clinicians working in rural British Columbia, Canada. In this study, it was found that 75% of the clinicians recruited from within the community (locals) agreed or strongly agreed that, overall, they were “very satisfied with their rural lifestyle”, compared to 55% of those recruited from outside the community (non-locals) [51].

### 3.2. Social Connection and Place Integration

Opportunities to meet people and develop social networks in the place they moved to was found to be important for retention for non-local health professionals across a number of studies [27,43,49]. In a focus group study with 30 allied health professionals working in rural New South Wales (NSW), Australia, the authors found that community engagement and personal relationships were powerful motivators to stay (or, in their absence, leave) [22]. Auer and Carson argued that forming place attachment is both a necessary and an essential human process for successful adjustment to a new environment [52]. Godwin et al., in their systematic review of dental practitioners’ rural work movements, identified that social factors were the most common influence on retention [44]. They also found that, if a dental practitioner felt lonely or isolated, or did not have a close support network, they would eventually leave their rural position, irrespective of how much money was on offer [44].

Almost all the non-locals in Cosgrave et al.’s study of factors affecting turnover intention of nursing and allied health professionals working in rural public sector community mental health services in NSW, Australia, reported feelings of alienation and social disconnection, especially in the first year of moving to a rural town [27]. The extent of this social isolation was found to be more intense for new graduates, people who were single, or people who had limited or no prior experience of rural “Aussie” (Australian) cultures [27]. This study also found that non-local newcomers who came with partners or relocated with family members were, to some extent, insulated from initial experiences of loneliness and social vulnerability [53]. Having partners and/or children was also found to broaden the social net from which an individual could make friends [53]. Relatedly, Hancock et al. argued that the integration of non-locals must be viewed through the lens of the entire family, and partners must be included in efforts to support integration [46].

Integration among non-locals was observed in several studies to be a gradual, stepped process [27,46,54]. The challenging initial adjustment period that Cosgrave et al. identified non-locals experiencing was also observed in other studies [47,49,52,54]. For example, Pierce discussed an “integration process” involving feelings of loss and displacement [54]. In Allan et al.’s retention study of pharmacists and social workers in rural NSW, Australia, “towns” were described as having individual characteristics and particular “personalities” that shaped the social activities on offer and the availability of local groups to join [55]. In this sense, the particular cultural context of a town was identified as affecting non-locals’ ability to fit in and form a sense of belonging, and their overall desire to live in the town. Several studies found that non-locals rarely made friends with locals, and most friendships were formed with other non-locals and, initially at least, there was particular reliance on work colleagues for social connection [43,47,52,53]. Cosgrave’s study argued that making friends with other non-locals increased the social vulnerability of non-locals, as this group generally had less place attachment to the town and were, thus, more likely than locals to leave [53].

### 3.3. Community Participation and Satisfaction

In many of the reviewed studies, satisfaction with community was linked to job satisfaction [56,57,58,59]. Penz et al. argued that this link is a phenomenon unique to rural contexts [58]. In her study, Pierce argued that the professional and personal identities of rural health professionals were inseparable [54]. Kelley et al.’s study of physicians working in rural practice in Ontario, Canada, found that professional dissatisfaction alone did not affect future practice intention; personal or family dissatisfaction was also required to induce the decision to leave [57]. The most common aspects found to influence community satisfaction across the literature were individuals’ sense of belonging to community, attachment to place, and, related to both of these, enjoyment of rural lifestyle. Aspects of sense of belonging included being known and greeted by name, able to participate in local groups and activities, and change things in their community environment [43,52,60].

Gillham and Ristevski’s study found that the non-locals most likely to leave were those whose main social connections continued to be based outside of the local area following relocation. This study also found that non-locals whose spouses had moved with them were more likely to stay than those whose spouses lived elsewhere [43]. Relatedly, Auer and Carson’s study found that health professionals who had children living with them were more likely to feel a sense of belonging than those who did not [52]. These authors discussed place attachment, including to the physical environment, as an important dimension to a sense of belonging [52]. Place attachment in rural areas was also found to develop through individuals’ involvement in outdoor recreational activities [49,52]. Furthermore, rural towns’ physical attractiveness and proximity to major capital or regional cities was found to increase duration of stay amongst non-locals [43,45,56,60].

Auer and Carson’s study of physicians working in the Northern Territory (NT, the Northern Territory is classified by the Australia Bureau of Statistics as mostly “very remote” or “remote”) found that most physicians were non-locals and nearly all expressed that their time living and working in the NT was a “temporary adventure” [52]. Participants spoke of living in the NT in terms of involving many lifestyle sacrifices (for example, a lack of suitable housing, distance to desirable services and facilities, lack of social connection, and limited spousal employment options). Thus, participants viewed their employment as time-limited, most commonly as a 1–2-year commitment [52]. May et al.’s study found that the climate and environmental amenities in coastal towns were the most important retention factors for medical specialists [60]. Conversely, Gillham and Ristevski’s study found that rural towns situated in or near industrial areas (for example, open-cut mines) reduced the lifestyle appeal of the area and did not feature as a motivational (pull) factor for recruitment or retention [43]. The rural lifestyle features identified as influencing retention included a laid-back and relaxed way of life, having more time to spend with family, experiencing less traffic, the smaller size of the community, greater housing affordability, a less materialistic way of life, and the place being a good environment for raising children [22,48,49,52].

### 3.4. Fulfillment of Life Aspirations

Pierce identified retention of non-locals as processes encompassing “coming here”, “being here”, and “staying here”, and argued that the transition for non-locals from “being here” to “staying here” involves a merging of personal identity with rural place. She found that among participants who were long-term stayers all expressed a deep satisfaction with their “identity-in-place” [54]. Manahan et al. also identified this merging of identity with place and its influence on retention in their study identifying the personal characteristics and experiences of allied health professionals who had worked long term in northern British Columbia, Canada [49]. Cosgrave et al. argued that the duration of stay for rurally-based health professionals was dependent on the extent to which the personal and professional needs of the health professional, and any significant others, were able to be met by the rural town and community in which they were living [27]. Hancock et al.’s study of rural-based physicians in California and Nevada, USA, investigating physicians’ practice location choices over the life course, observed that, as well as desiring place and community integration, living a happy and satisfying life was an important motivation [46]. The authors identified that, once more basic psychological needs had been met, physicians had increasingly complex desires and motivations. While the limitations of Maslow’s hierarchy of needs model were acknowledged, it was argued that such a framework is still useful for understanding physicians’ motivations [46]. The choice to remain in rural practice for an extended period of time was generally found to be made by those health professionals who were in their middle years, partnered, and/or raising a family, especially if the children were preschool or primary school aged [27,45,57,61]. Cosgrave et al. argued that life stage, rather than rural origin, was the major determinant of turnover for both locals and non-locals, and that those in early adulthood (usually in their early-to-mid 20s) were the most likely to leave within a couple of years of working, regardless of background [27]. This group of health professionals usually left to travel, seek new adventures, extend social networks, or, for non-locals, return home [22,27]. The authors also observed that place and community integration increased over time and those non-locals who had stayed beyond three years were generally in middle adulthood and as likely to stay as locals [27]. Limited local secondary school options were identified as a major reason why health professionals with children entering secondary school education decided to leave their rural positions [22,52]. Other life-course events such as retirement were also found to influence the duration of stay among non-locals [52].

## 4. Discussion

Increasingly over the last decade, the importance of place-based social processes for rural health workforce retention was identified in the literature. However, many retention research studies to date lacked analytical depth; only two published studies [27,46] proposed a conceptual framework that moved beyond a simple push and pull understanding. Furthermore, much of the existing research focused on doctors [46,52,57,60], dentists [44,48], or pharmacists [55,61]. These are generally considered high-status health professions [62]. Thus, these professions are the targets of political attention and government-initiated and/or -supported retention strategies and/or policies aimed at individual professionals and their family units. These individuals and their families can be more easily directly supported by local health services, businesses, and/or rural communities. Consequently, much of the literature pertaining to the retention of these health professionals focused on identifying individual-level needs and how they can be met. In comparison, retention of allied health and nursing professionals needs to be considered at a workforce level. That is, strategies for recruiting and retaining allied health and nursing professionals cannot exclusively target individual practitioners; allied health and nursing recruitment and retention strategies need to also focus at the workforce level. This is especially important in rural contexts, where the bulk of the health workforce is employed in the public sector [9].

Identifying and implementing effective solutions that can respond to needs at an individual and workforce level is a much more challenging endeavor. To date, there was little progress in responding to the place-based social processes affecting retention of a rural health workforce, despite emerging recognition of their importance. In fact, the research reviewed in this paper suggests that place-based social processes are the most central to retention, yet remain the least explored and addressed dimension. More research on the place-based social dimensions influencing retention is, thus, urgently needed. However, future research must go beyond the identification of contributing factors and offer robust understandings of the social determinants of rural retention, so that responsive policy initiatives can be developed and trialed. In short, the social determinants of retention need to be conceptualized in ways that reflect the complexities of and variances involved in the processes and contexts at play.

To progress the rural workforce retention research agenda, more clarity and agreement is needed concerning the common language used in rural health retention research (for example, turnover, retention, rural, regional and remote, sense of connection and/or place, community, belongingness, outsiders, newcomers, locals, non-locals, career, and life stage), so that stakeholders can be clear about the specific issues or phenomena under interrogation. The research so far undertaken also suggests an urgent need for more nuanced conceptualizations of retention that reflect, in particular, a life-stage perspective on likely duration of stay. Such conceptualizations need to be effectively disseminated to and taken up by managers and executive staff in rural health services so that internal retention goals and strategies can be appropriately developed. From this point, stronger conceptual frameworks that take into account, deconstruct, and respond to the social determinants of rural health workforce retention can be built and applied in the field.

Figure 2 synthesizes the findings from the current rural health workforce research reviewed in this paper to provide a useful starting point for constructing such frameworks. It is our contention that future conceptual frameworks need to clearly identify those place-based social processes that health services and/or rural communities can influence, that is, those that are amenable to change (for example, social connection) as opposed to those that are not (for example, life-stage influences and life-course events). The purpose of identifying place-based social processes that are amenable/not amenable to change is to build a depth of understanding about individual/family unit contexts and social situations that heighten turnover risk, so that appropriate responses and supports can be designed and implemented. It is not to limit recruitment to those who are most likely to stay (for example, health professionals in middle adulthood, partnered, and/or with children who are primary school aged). It is always important for access, equity, and inclusion to have a diverse and inclusive workforce reflecting the community it serves.

### Study Limitations

This scoping review focused on the social determinants of retention, and established that place-based social processes are an important influence on rural health workforce retention, both positively and negatively. However, it is important to acknowledge that rural health workforce retention is a complex phenomenon and is influenced by a broad range of issues. These include the funding and policy setting for both rural health services and the various health professions, work roles, team and organizational culture, and access to professional development and career-building opportunities. Undertaking further research to better understand place-based social processes will provide a more complex and complete understanding of rural health workforce retention. Other more minor limitations of this study were previously noted in this section.

## 5. Conclusions

This review described the range of literature that addresses the influences of place-based social processes on the retention of allied health, nursing, and/or medical professionals in rural communities. There is sufficient evidence across the existing literature to consider place-based social processes to be important influencers affecting the retention of rural health workforces. Thus, further research focused on the social dimensions of retention is needed, and comprehensive conceptual frameworks that consider and enable thorough exploration of these determinants should be developed. The authors suggest that, in the future development of research in this area, critical and focused attention should be given to identifying those place-based social processes that are amenable to change (and those that are not). This will enable the generation of responsive, evidence-informed change initiatives for application in the field. Such future research will generate new insight into how rural health professionals can be recruited and, importantly, retained. This research will have particular resonance in the fields of rural allied health and nursing, where there are currently the most severe shortages, by allowing for retention to be conceptualized and approached holistically at a workforce level. It is hoped that, with the translation of this research into practice, more stable and adequate health workforces can be secured for rural communities, and a reduction can be achieved in current health disparities between those living in rural and metropolitan areas.

## Figures and Tables

**Figure 1 ijerph-16-00314-f001:**
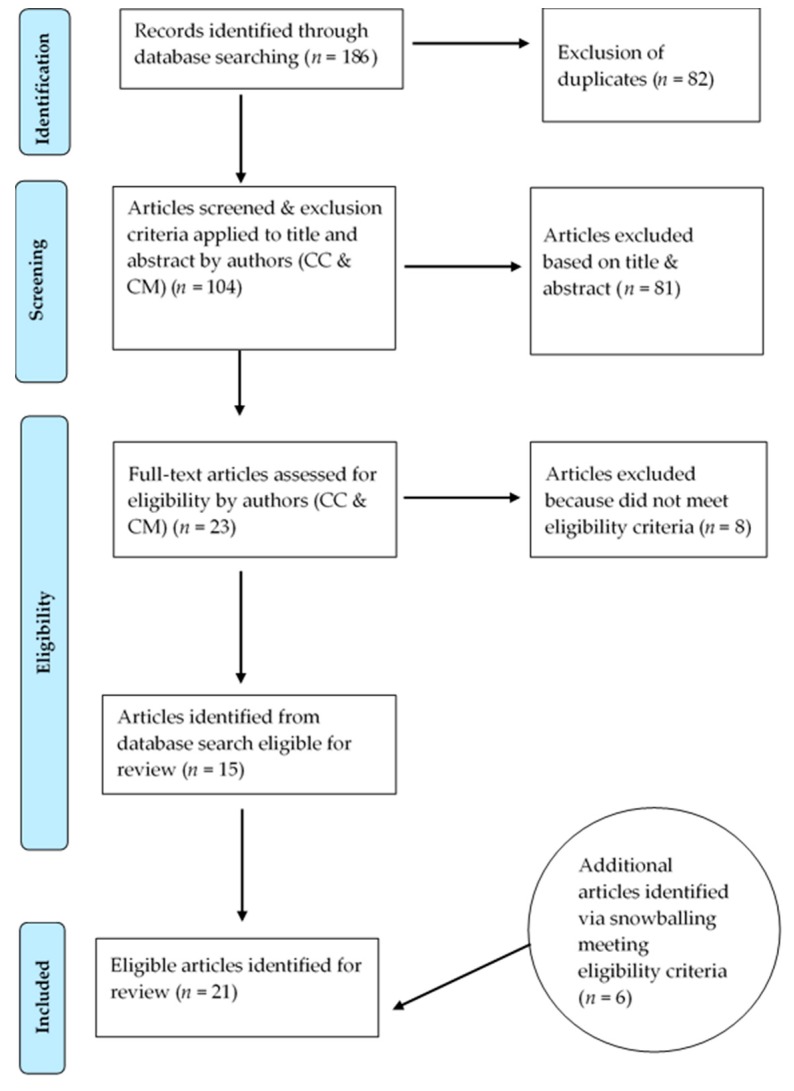
Flowchart of scoping review process.

**Figure 2 ijerph-16-00314-f002:**
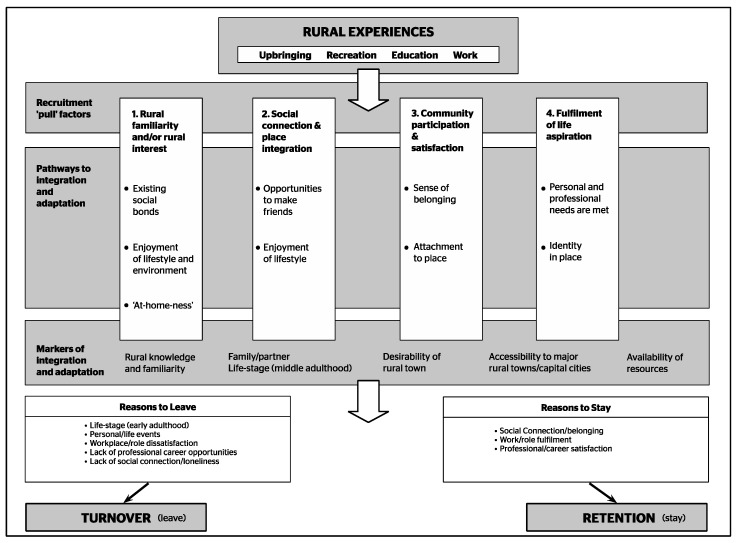
Conceptual framework of the social determinants of rural health workforce retention. Based on Hancock et al. [46] (with permission).

**Table 1 ijerph-16-00314-t001:** Characteristics of included literature.

Characteristics	Number of References
Types of paper/study	
Qualitative studies	11
Quantitative studies	8
Mixed methods	1
Literature reviews	1
Total	21
Location/region of focus	
Australia—Northern Territory	2
Australia—New South Wales	6
Australia—Victoria	2
Australia—national rural	1
Total	11
Canada—British Columbia	4
Canada—Ontario	1
Canada—rural areas	3
Total	8
USA—California and Nevada rural areas	1
USA—National rural areas	1
Total	2
Grand total	21

**Table 2 ijerph-16-00314-t002:** Themes and categories identified in the reviewed literature.

Themes	Categories	Article Number as per Table 3
Rural familiarity and/or interest	Rural origin and pre-existing social networks have a strong “pull” influence	5, 6, 8, 9, 10, 15, 18
	Rural interest (prior exposure, adventurous inclination, interest in outdoor recreational pursuits, and personal circumstances (e.g., a partner)) also influence recruitment	7, 14
	Social connection and place integration are a necessary human process	2
Social connection and place integration	Challenging initial adjustment experience for non-locals	2, 3, 15, 18, 20
	Partner/family living with person reduces isolation and extends social network	3
	Characteristics/culture of town can support or obstruct connection/integration of non-locals	1
	Social connections are usually made with other non-locals who often rely on work colleagues for social connection, especially initially	2, 3, 6, 18
Community participation and satisfaction	Rural health positions job satisfaction and community satisfaction are linked	11, 13, 17, 19
	Community satisfaction involves sense of belonging to the community, attachment to place, and enjoyment of rural lifestyle	2, 6, 15, 16
	Town’s geographical attractiveness and its proximity to major capital or regional cities increases duration of stay	5, 6, 11, 16
	Rural lifestyle	2, 9, 12, 15
Fulfillment of life aspirations	Long-term stay requires merging of personal identity with place	15, 20
	Over medium–long term (3+ years) life stage, not rural origin, is major determinant of retention	4
	Retention influenced by whether town/community can meet the future personal development needs of the health professional and their significant others	4
	Long duration stays most common among health professionals (local and non-locals) in their middle years, partnered, and/or involved in raising a family, especially if children preschool or primary school aged	4, 5, 13, 21
	Life-course events (e.g., children’s secondary school) push factors	2, 12,
	Health professionals in early adulthood have generally short duration of stay (less than 2 years); locals leave position to travel, for adventure, and extend social network; non-locals leave either for these reasons or to return home	4, 12

**Table 3 ijerph-16-00314-t003:** Methodological details for and relevant findings in reviewed literature.

No.	Author (Year)	Methods	Study Population and Response Rate	Focus	Relevant Findings
1	Allan et al. (2008) [55], New South Wales (NSW), Australia	Qualitative interviews	Accidental sampling pharmacists and social workers working in towns of less than 5000 residents 11 participants (5 social workers and 6 pharmacists)	Rewards and barriers experienced in professional, personal, and social lives and plans for the future	Each town has individual characteristics and a unique culture influencing social activities and groups on offer, which affects new, non-local staff’s ability to fit in and belong A town’s character can make it desirable/undesirable to live in Retention is a complex combination of local context, professional role, and personal relationship
2	Auer and Carson (2010) [52], Northern Territory (NT), Australia	Qualitative interviews	19 participants General practitioners (GPs) 14 GPs stay length less than 3 years and 5 GPs stay length over 3 years	Place attachment experiences of GPs who moved to NT to work Why they took a job in the NT, how they have come to feel about community, and the factors influencing their decision to stay or leave	All GPs new to the area/town desire place attachment The work and the physical environment are major factors in place attachment Many lifestyle sacrifices experienced in remote areas, resulting in employment duration usually being limited to one or two years Life-stage events for either the GP and/or their family members (e.g., child’s secondary schooling) most often trigger the decision to leave While stay in NT is nearly always temporary, most GPs made a deliberate choice to enjoy the experience Social connection commonly leveraged through work relationships and friendships mostly made with other non-locals Most GPs achieved a sense of place attachment (i.e., were greeted by name and participated in local groups) GPs with children the most likely to form social networks
3	Cosgrave (2015) [53], NSW, Australia	Qualitative interviews	26 nursing and allied health professionals working in public community health services in rural and remote areas	Professional and personal factors impacting turnover intention of early-career, rural-based community mental health professionals in their first few years of working	Non-local newcomers usually make friends with other non-locals Non-local newcomers who moved with partner and/or children had a broader social net for making friends
4	Cosgrave et al. (2018) [27], NSW, Australia				Non-local newcomers experience alienation and social disconnection, especially in the first year of living in the town Feelings of belonging increase incrementally the longer non-locals live in town, similar to locals after 3 years Rural background has limited influence on turnover intention, life stage is the primary determinant
5	Gallego et al. (2015) [45], NSW, Australia	Quantitative survey	429 allied health professionals (AHPs) working in western NSW emailed, and 218 completed online survey Estimated response rate was 51%	Characteristics of AHPs working with people with disabilities	AHPs with dependent children less likely than those without dependent children to cease working within 5 years The migration intentions of people in rural areas are complex and variable and not only driven by individual characteristics (e.g., age, marital status), but community attachment and amenities
6	Gillham and Ristevski (2007) [43], Victoria (VIC), Australia	Qualitative interviews	43 allied health professional (AHP) participants from two rural services (8 students, 18 current staff, 7 managers, and 10 former staff)	Recruitment and retention issues affecting AHPs	Social networks are an important recruitment factor; student participants keen to take their first job close to home for reasons of social support Non-local staff keen to make new friends; socializing mostly occurred with/through work colleagues, they found it difficult to get to know “locals” Staff who stayed usually made social connections outside the workplace or already had connections in the community If main social networks remained outside the area, person tended to move on faster The location of AHPs’ partners had a significant impact on staff retention; those with a partner in the local area more likely to stay than those with one based elsewhere Lifestyle factors did not influence recruitment or retention, contrasting with other rural AHP workforce studies where lifestyle was important for attracting and retaining AHPs The two services were situated in a fairly industrial area, with open-cut mines and power stations, which likely reduced lifestyle appeal
7	Gillespie and Redivo (2012) [51], British Columbia (BC), Canada	Mixed methods Results from online questionnaire	44 child and youth mental health clinician respondents Response rate 48%	Factors impacting the recruitment and retention of child and youth mental health clinicians living and working in rural localities	Clinicians most likely to be satisfied with rural lifestyle and most likely to find their practice rewarding were those recruited from within the community—75% of locals agreed or strongly agreed that, overall, they were “very satisfied with their rural lifestyle”, compared to 55% of non-locals
8	Godwin et al. (2014) [44], Australia	Literature review (systematic)	16 articles met the inclusion criteria	Factors influencing dental practitioners’ decisions to come to, stay, and leave rural and remote areas	Most influential long-term rural practice retention factors were personal/social Positive and negative motivational factors influenced the decision to work in, remain working in, or leave rural practice. These included personal support networks, successful integration into the community, and enjoyment of rural lifestyle for both the individual and family members When individuals were lonely or isolated and without a close support network, they left, irrespective of how much money was offered
9	Hall et al. (2007) [48], NT, Australia	Quantitative survey	73 respondents—42% response rate After eligibility criteria applied, 43 respondents	Factors influencing dental practitioners to move to/from NT	Social, not work-related factors were the most important in attracting/retaining dental practitioners Those choosing to stay were more likely to be involved in social activities/groups, own their own home, enjoy small size of towns/cities, and be content with the rural lifestyle
10	Hancock et al. (2009) [46], California and Nevada, USA	Qualitative interviews	22 participants—primary care physicians working in remote areas of California and Nevada	Practice location choice over the life course of primary care physicians working in rural communities	Physicians drawn to rural practice because of 4 key factors: familiarity, community involvement, place integration, and supportive of achieving self-actualization goals Physicians from a rural background often chose to work in their home town or a town nearby or place with a similar geography and population size Place integration was a process and needs to be viewed through the lens of the entire family Partners needed to be included in efforts to facilitate community integration and connection
11	Henderson-Bektus and MacLeod (2004) [56], BC, Canada	Quantitative survey	124 Public Health Nurses (PHNs) working in rural British Columbia in 37 rural sites Of the 124 respondents, 68 were from rural communities, and 56 were from small urban communities Response rate—76% (164 eligible PHNs)	Examined job and community satisfaction and how it relates to decision to stay or leave current rural employment	Most satisfying community aspects were friendly community, having friends, and size of community Overall, 63% were satisfied to very satisfied with their community Least satisfying community aspect was distance from a major center Workers’ demographics, personal circumstances, and opportunities all found to affect retention regardless of the level of job and/or community satisfaction
12	Keane et al. (2012) [22], NSW, Australia	Qualitative focus groups	Purposive sampling 6 focus groups, 30 allied health participants (AHPs) conducted across rural NSW	Identified aspects of recruitment and retention affecting allied health professionals working in rural locations	Community engagement and personal relationships found to be powerful motivators for retention Many AHP participants either grew up rurally or were attracted to a rural lifestyle Those who were parents felt rural towns were good places to raise their children “Push” factors in the personal domain included insufficient community infrastructure such as transportation, secondary schools, access to shops, and spousal employment Access to adequate accommodation was a concern in more remote regions Younger AHP professionals left early in their employment for travel or adventure, or to find a better peer social environment
13	Kelley et al. (2008) [57], Ontario, Canada	Quantitative survey	201 physicians working in north west Ontario 61.3% response rate	Factors affecting future practice intentions of physicians practicing in rural and underserviced areas	Physicians more likely to stay in practice if they felt a sense of belonging Physicians who intended to stay were also more dissatisfied with professional aspects of rural medicine, suggesting professional factors alone did not influence future practice intention but also needed personal/family dissatisfaction
14	Kulig et al. (2009) [50], rural Canada	Quantitative survey Results from a thematic analysis on open ended questions	3933 registered nurses (RNs) responded 68% response rate Analysis on a subset of 3331 RNs working in rural and remote areas	Community satisfaction and attachment among RNs working in rural and remote areas	Two types of community attachment among rural RNs: “going home” or “becoming home” Community satisfaction for rural RNs identified as having two types: “being rural” or “becoming rural”
15	Manahan et al. (2009) [49], BC, Canada	Qualitative interviews	22 allied health professional (AHPs) participants working in northern BC	Personal characteristics and experiences of AHPs who worked long-term in Northern BC	Community satisfaction found to be important influence on AHPs’ decision to stay Factors identified as influencing retention: getting involved in the community, appreciating the physical environment, the recreation activities available, and the quality of life on offer Retention influenced by the extent of feelings of belonging to the community Friendliness of the community major factor in the decision to stay Most participants did not have friends in the region on arrival, but developed close friendships over time, which influenced them to stay long term Strongest factor influencing retention was family and wanting to spend time with family members, especially children and grandchildren
16	May et al. (2017) [60], NSW, Australia	Qualitative interviews	62 medical resident specialists working in 4 regional centers (2 coastal and 2 inland)	Recruitment and retention factors important to medical specialists’ location decision-making	Sense of community was highly rated and partner employment moderately important as retention factors Sense of community was valued by many respondents and described as a “sense of being known” and “being able to change things” Specialists residing in coastal places rated location factors at a much higher level for both recruitment and retention Notably, climate and environmental amenity eclipsed even the professional factors for importance in retention for coastal specialists
17	Muus et al. (1998) [59], rural USA	Quantitative survey	A random sample of 1263 physician assistants (PAs) practicing in rural areas of the USA Average response rate 64.85%	A statistical model to measure job satisfaction of Pas Model tested for its ability to predict levels of job satisfaction using multiple regression analysis	Consistent with most studies, satisfaction with community was found to be one of the strongest predictors of the model Results indicated that PAs who were happy with the community in which they practiced were likely to be satisfied with their job
18	O’Toole et al. (2010) [47], VIC, Australia	Qualitative interviews	32 allied health professionals who had left a health position in rural Victoria	Working experiences and reasons for resignation among rural-based allied health professionals	Personal reasons for taking up a position in a rural area accounted for most of the responses from participants, with most responses relating to being close to family and friends
19	Penz et al. (2008) [58], rural Canada	Quantitative survey	944 rural and remote registered nurses working in acute care Part of a larger nationwide study Response rate 68%	Relationship between the individual, workplace, and community characteristics as predictors of job satisfaction among acute care RNs working in rural hospitals	Satisfaction with home community was one of 4 significant predictors of job satisfaction (other 3 work-related) Concept of community satisfaction as an aspect of job satisfaction unique to rural-based health workforce Some participants found their position assisted them to connect into the community, described feeling an affinity with their peers, which helped them feel more socially connected to the organization and the community Some participants indicated that, because of living in a rural area, they felt socially isolated and had less affinity with their community
20	Pierce (2017) [54], BC, Canada	Qualitative interviews	Four social work professionals—3 from large urban centers with no prior experience living or working in isolated communities, 1 had returned to her/his “home” community—all intending to stay	Influence of place and processes of place attachment on retention of health workers in remote locales	Feelings of loss and displacement from previous places impacted on health professionals’ willingness and ability to bond to the current place Initially viewed their situation as something temporary, to be endured, and had no plans to settle The transition from “being here” to “staying here” involved a merging of identity and place—each participant now experienced a deep satisfaction with their identity-in-place Professional and personal identities inseparable for practitioners in rural and remote settings
21	Woodend et al. (2004) [61], rural Canada	Quantitative survey	1019 rural pharmacists Response rate 40%	Predicting intent to remain in rural practice	Pharmacists’ satisfaction with both professional and personal aspects of living and working in a rural community significantly associated with their intention to remain in practice in that community Younger age, being married, and the presence of children at home were associated with rural pharmacists’ intention to remain working in the community; only age remained significant when these predictors were considered simultaneously in a univariate analysis Pharmacists in both their middle years of practice and middle age more likely to remain

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
