# Peer review of "Social Determinants of Rural Health Workforce Retention: A Scoping Review"

_ijerph, 2019, doi:10.3390/ijerph16030314_

Round 1

Reviewer 1 Report

I think this is a  very useful piece of work to underpin understanding in this important area.

Several areas that I was surprised not to see mentioned, and of which I have knowledge are:

1: Professionals with children who have high educational aspirations for those children not available in regional or remote areas and necessitating older children having to go to city colleges as borders.

2: Unlike the city, the inability to 'get away' from patients i.e. in simple shopping outings, etc. This can be particularly challenging in small communities and with 'difficult' patients.(Notably Dental patients)

3: Lack of relatively  'sophisticated' entertainment opportunities.

4: 'Always' on call. 

Of course I understand that several of these issues are absolutely endemic and relatively unchangeable.

It would be really good to see any follow up work which might address solution or intervention for these issues.

Reviewer 2 Report

A relevant and contemporary paper of broad interest to the readership of this journal. This topic is of importance in service design and policy development and is a worthy target for a scoping review.

The abstract is concise and reflects the focus of the review and the findings in a clear manner. The final sentences are a bit of an overreach based on the evidence in the paper and could be toned down.

The introduction covers a diverse range of issues linked with rural health workforce, outlining important concepts and background.

Page 2 line 50 the point about greater reliance on the public sector requires a reference to support this point.

Page 2 line 65 & 66 is a strong point and requires a reference to support this specific point. My view is that there are a number of issues around the lack of allied health professionals in rural and remote areas.  

There is not a clear statement about place based and social processes in the background which leaves the reader unclear as to where the concepts arise from in relation to rural and remote health workforce. I would appreciate some reference to what is meant by these terms supported by references. The concepts make sense and I can follow your argument that they impact on rural and remote health workers, but the argument is not well supported in the introduction. The final paragraph of the introduction begins with “Given the important role of place-based social processes…” does not follow from the information in the preceding paragraphs.

The materials and methods section is clear and includes the research questions in a concise manner.

The definitions of key terms is helpful and clarifies the vocabulary of the paper. However, I am not sure that the use of the terms in lines 134 & 135 interchangeably is helpful as they cover quite different concepts.

The search terms are clearly outlined along with the search process.

The inclusion and exclusion criteria section does not provide enough detail on what is meant by place based social processes as this has not been previously described. The exclusion criteria are appropriate and clearly articulated.

The initial results paragraph is well structured and introduces the themes and data in a clear manner. The use of tables and flow charts aids the readers understanding of the data. Table 2 is a little confusing with the references to the articles not aligning with the numbers allocated to the papers in Table 3, but aligned with the numbering in the reference list. This could be made clearer.

The first two themes are well described, however, the writing could be further integrated with less use of the authors name at the beginning of the sentence which interferes with the flow of ideas. Further integration of the concepts revealed by the different studies that fit under that theme would help the reader understand the content of the articles in a more synthesised manner. The third and fourth themes demonstrate better flow and synthesis of the ideas and are much easier to read. 

The discussion includes a comprehensive integration of the content of the results, however, I am concerned that there is an over-reach in the impact of place based social processes. This is the focus of this research, but it must be placed in the context of other factors that are present in the complexity of rural and remote allied health workforce, such as policy settings, funding and work roles. Research on the place based social processes is needed, but the tone of this finding could be moderated and contextualised. The framework in Figure 2 is a great contribution to the synthesis of this body of work and is clearly presented.

The limitations section could include an acknowledgement of the broader context of retention factors that should be considered with social place based processes to generate a more complex and complete understanding of allied health workforce in remote areas.

The conclusion that place based social processes is one of the most important factors in retention is a bit of an over reach. It is an important factor, but it is difficult to support the contention that it is on of the most important factors. The conclusion that policy responses are required is an important finding. The point on line 375 about workforce shortages being set to rise further needs a reference. The aspiration of equity and improvement in rural health workforces is appropriate and will be aided by the contribution of this research.
